# Risk assessment of COVID-19 epidemic resurgence in relation to SARS-CoV-2 variants and vaccination passes

Tyll Krueger[1,14], Krzysztof Gogolewski [2,14], Marcin Bodych[1,14], Anna Gambin [2], Giulia Giordano [3], Sarah Cuschieri[4], Thomas Czypionka[5,6], Matjaz Perc [7,8,9,10], Elena Petelos [11,12], Magdalena Rosińska [13] & Ewa Szczurek [2✉]

**Abstract** The introduction of COVID-19 vaccination passes (VPs) by many countries coincided with the Delta variant fast becoming dominant across Europe. A thorough assessment of their impact on epidemic dynamics is still lacking. Here, we propose the VAP-SIRS model that considers possibly lower restrictions for the VP holders than for the rest of the population, imperfect vaccination effectiveness against infection, rates of (re-)vaccination and waning immunity, fraction of never-vaccinated, and the increased transmissibility of the Delta variant. Some predicted epidemic scenarios for realistic parameter values yield new COVID-19 infection waves within two years, and high daily case numbers in the endemic state, even without introducing VPs and granting more freedom to their holders. Still, suitable adaptive policies can avoid unfavorable outcomes. While VP holders could initially be allowed more freedom, the lack of full vaccine effectiveness and increased transmissibility will require accelerated (re-)vaccination, wide-spread immunity surveillance, and/or minimal long-term common restrictions.

**Plain language summary**

Assessing the impact of vaccines, other public health measures, and declining immunity on SARS-CoV-2 control is challenging. This is particularly true in the context of vaccination passes, whereby vaccinated individuals have more freedom of making contacts than unvaccinated ones. Here, we use a mathematical model to simulate various scenarios and investigate the likelihood of containing COVID-19 outbreaks in example European countries. We demonstrate that both Alpha and Delta SARS-CoV-2 variants inevitably lead to recurring outbreaks when measures are lifted for vaccination pass holders. High re-vaccination rates and a lowered fraction of the unvaccinated population increase the benefit of vaccination passes. These observations are important for policy making, highlighting the need for continued vigilance, even where the epidemic is under control, especially when new variants of concern emerge.

[1] Faculty of Electronics, Department of Control Systems and Mechatronics, Wroclaw University of Science and Technology, Wroclaw, Poland. [2] Faculty of Mathematics, Informatics and Mechanics, University of Warsaw, Warsaw, Poland. [3] Department of Industrial Engineering, University of Trento, Trento, Italy. [4] Department of Anatomy, Faculty of Medicine and Surgery, University of Malta, Msida, Malta. [5] Institute for Advanced Studies, Josefstädterstraße 39, 1080 Vienna, Austria. [6] London School of Economics and Political Science, Houghton Street, WC2A 2AE London, UK. [7] Faculty of Natural Sciences and Mathematics, University of Maribor, Koroška cesta 160, 2000 Maribor, Slovenia. [8] Complexity Science Hub Vienna, Josefstädterstraße 39, 1080 Vienna, Austria. [9] Department of Medical Research, China Medical University Hospital, China Medical University, Taichung 404332, Taiwan. [10] Alma Mater Europaea, Slovenska ulica 17, 2000 Maribor, Slovenia. [11] Clinic of Social and Family Medicine, Faculty of Medicine, University of Crete, Heraklion, Greece. [12] Department of Health Services Research, CAPHRI-Care and Public Health Research Institute, Maastricht University, Maastricht, The Netherlands. [13] Department of Infectious Disease Epidemiology and Surveillance, National Institute of Public Health, Warsaw, Poland. [14] These authors contributed equally: Tyll Krueger, Krzysztof Gogolewski and Marcin Bodych. ✉email: szczurek@mimuw.edu.pl

n the past, governments have required proof of vaccination for travel, with yellow fever being the best-known example, and the only disease for which a certificate is needed as a pre-condition of entry to a country in compliance to the International Health Regulations[1]. However, the idea that proof of vaccination will become a prerequisite for crossing borders or to enter facilities, visit businesses premises, participate in events, and generally enjoy more freedom, only arose in the context of combatting the COVID-19 epidemic. Despite technical challenges, scientific uncertainties, and ethical and legal dilemmas, the idea of VPs, i.e., documents issued on the basis of vaccination status, received unprecedented attention[2–4]. The Commission of the European Union (EU), in an effort to ensure a uniform pan-European approach, as similar initiatives for VPs were emerging at national level, put forth a proposal for a framework of issuing, verifying and accepting interoperable vaccination certificates to be implemented across the EU[4], along with a corresponding proposal for third-country nationals residing in the EU[5]. The proposal, in its amended form, for the 'Digital COVID Certificates' (DCCs), took effect on July 1, 2021. Many consider the EU DCCs, and other forms of VPs in general, as tools to restore people's freedoms and increase well-being, whilst allowing economies to reopen. Finally, even without VPs, vaccinations alone may result in less stringent behavior. Those vaccinated may feel more secure and restrict themselves less from contacts they would refrain from when not being vaccinated.

The introduction of VPs and consequent changes in behavior coincided with the emergence of new variants of concern of the virus[6]. Notably, the Delta variant (B.1.617.2) was detected in many countries across Europe, causing a resurgence of COVID-19 in the United Kingdom at a startling pace[7,8]. Delta was estimated to be 50% more transmissible than the Alpha variant (B.1.1.7), already estimated to be 50% more transmissible than the previously dominant strains[9–11].

Evidence indicates vaccine effectiveness can greatly vary[12,13] and it may be compromised due to escape variants[14] and waning immunity[15–18]. Preliminary data from several countries indicate reduced vaccine effectiveness against the infection with the Delta variant compared to the Alpha variant[19–21], even as low as 64% for the Comirnaty (Pfizer-BioNTech) vaccine according to data from Israel[22]. Emerging evidence suggests that the vaccines are effective in preventing serious illness and hospitalization[11,20,21].

Still, avoiding another COVID-19 infection resurgence remains a valid and potentially attainable goal[23]. Immunity against both infection and hospitalization wanes over time[15,18,24–28]. An estimated 10% of COVID-19 infections will have long-term sequelae (long COVID), posing an increasing threat to national health systems[29,30]. Finally, large numbers of infected create a large pool of virus hosts, resulting in more replications of the virus and higher chances of emergence of mutations conferring evolutionary advantage, including increased transmissibility and antigenicity. To detect the emerging variants, wide-spread surveillance of genetic and antigenic changes in the virus population has to be conducted, together with experiments elucidating their phenotypic implications[31]. Such needed comprehensive surveillance and experiments may become stalled for a large population of infected. Given these circumstances, it is critically important to understand the impact of key risk factors such as: vaccine ineffectiveness against infection, slow vaccination rate, waning immunity, fraction of individuals in the population who will never become vaccinated, and finally the levels of restrictions, on infection dynamics. Not being aware of the risks and their consequences, and a false sense of security, including when approaching higher vaccination coverage, may result in policy-makers opting to select suboptimal levels of restrictions.

Various models were developed to inform vaccination strategies[32–40]. One such effort indicates lower vaccine effectiveness coupled with an increase in social contact among those vaccinated (behavioral compensation) may undermine vaccination effects, even without considering immunity waning[41,42]. Scenarios for the post-vaccination era were also considered by Sandmann and colleagues (2021), finding that under realistic scenarios periodic epidemics are likely[43]. So far, there has been no model to focus on the medium- and long-term impact of relaxing restrictions for VP holders, with due consideration to vaccine effectiveness, durability of response, and vaccine hesitancy, especially in the context of the increased trans-missibility of the Delta variant. Given the implementation of the EU DCC, and emerging heterogeneous measures on utilizing the VPs for different purposes at national level by establishing different levels of freedom for VP holders in terms of accessing premises, facilities, traveling within a country, etc., it is important to examine the broad parameters determining how to optimize the implementation of measures such as the EU DCC and other VPs.

To address these needs, we propose a mathematical model called VAP-SIRS, which accounts for key parameters that impact the effective reproduction number of the virus, and consequently, infection dynamics: vaccination effectiveness, rates of (re-)vaccination and waning immunity, and the differences between SARS-CoV-2 variants. We perform comprehensive analysis for different levels of restrictions for VP holders and the rest of the population, for various realistic setups of these key parameters, including the different effectivenesses of the Comirnaty and Vaxzevria vaccines on the Delta and Alpha variants, as well as fractions of never vaccinated in the United Kingdom and France. The model predicts the impact of restrictions for VP holders and the rest of the population on epidemic thresholds for various parameter settings, and delivers a systematic framework to assess policy making. VAP-SIRS predicts a possible infection resurgence despite vaccinations. The resurgence is due to the lowered levels of restrictions for the VP holders compared to the rest of the population, while for some fraction of those VP holders the vaccine was ineffective and for the others the immunity may wane before they become re-vaccinated. A thorough analysis of our model identifies the complete set of potential scenarios for the COVID-19 epidemic depending on the restrictions imposed on VP holders and the rest of the population. For these scenarios, we estimate daily infection as well as hospitalization numbers and identify flexible measures to avoid epidemic resurgence. In particular, we derive the minimum common restriction level for the VP holders and the rest of the population, which can keep the epidemic subcritical in the long-term. Finally, we estimate the social benefit of VPs and find its strong dependence on (re-)vaccination rates.

## Methods

**Mathematical model.** We introduce VAP-SIRS (VAccination Passes in Susceptible-Infectious-Recovered-Susceptible model), as an extension to the classical SIRS model[44] (Fig. 1a). The population is divided into two subpopulations: those who are not vaccinated ($S$, $I$, $R$) and those who got vaccinated at least once ($S_V$, $I_V$, $R_V$, $V$). We assume that the group of non-vaccinated susceptible individuals $S$ (and, similarly, infected $I$ and recovered $R$) is divided into two subgroups: $S_N$ and $S_D$. The $S_N$ compartment contains such susceptible who will eventually be vaccinated, while those in $S_D$ will not.

The $S_N$ population is vaccinated with rate $v$ and effectiveness $a$. Consequently, the individuals from the $S_N$ group populate the vaccinated group $V$ with rate $av$. The individuals in $V$ are considered immune, and we assume that immunization prevents them both from getting infected and infecting others. The $S_V$ compartment is composed of $S_1$ and $S_2$ (and, similarly, vaccinated infected $I_V$ consists of $I_1$ and $I_2$). Due to vaccine ineffectiveness,

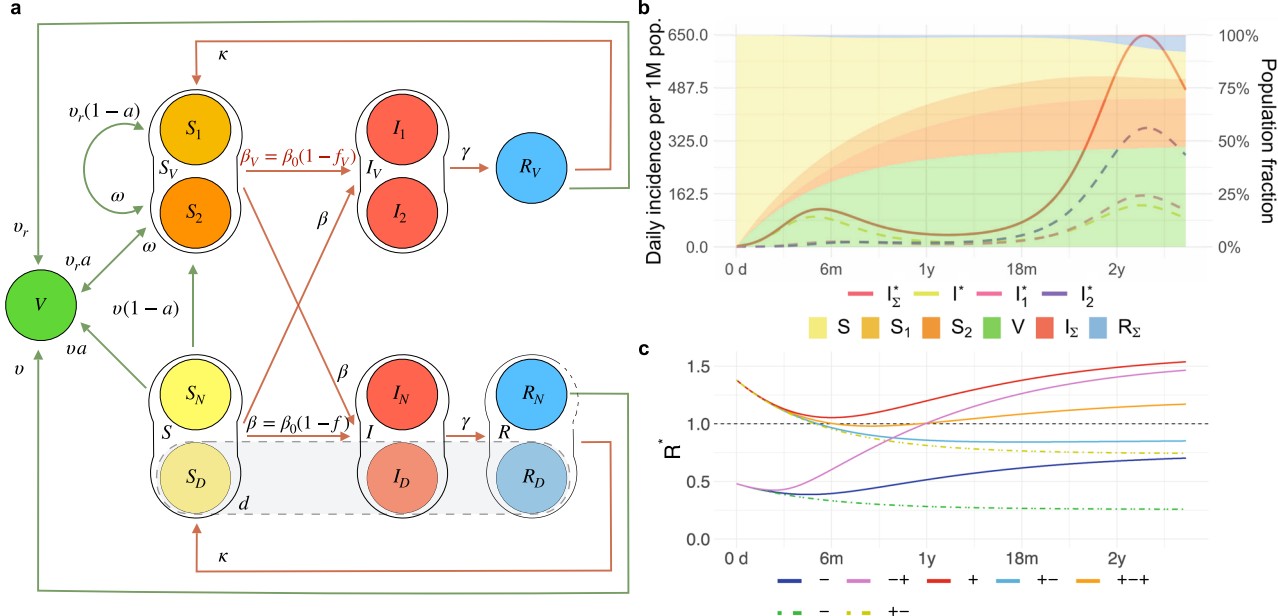

**Fig. 1 The VAP-SIRS model and its predicted scenarios. a** Graphical scheme of the VAP-SIRS model. **b, c** Predicted scenarios for the reference setup for the Delta variant, with vaccine effectiveness $a = 0.79$ (corresponding to the effectiveness of the Comirnaty vaccine against infection with the Delta variant), slow (re-)vaccination rate ($v = v_r = 0.004$; typical for many European countries), slow immunity waning $\omega = 0.002$, low fraction of never-vaccinated ($d = 0.12$; corresponding to the fraction in the United Kingdom) and proportional mixing (see Methods). **b** Color curves: Timeline of daily incidence per 1 million inhabitants in different infected compartments for the combination of restrictions $f = 0.77$ and $f_v = 0.55$. A variable with the asterisk (*) indicates that we consider a daily incidence over the corresponding variable. The dashed lines describe infected who are: non-vaccinated ($I^*$, yellow), vaccinated who did not gained immunity ($I_1^*$, pink), and vaccinated who already lost immunity ($I_2^*$, purple). By $I_\Sigma^*$ (red, solid line) we mean the sum of all daily infected ($I_D^* + I_N^* + I_1^* + I_2^*$). Color bands: Muller plot of the population structure (the width of the color band in the y axis) as a function of time (x axis) for the same parameter settings. Colors correspond to specific subpopulations: non-vaccinated susceptible ($S$, yellow), vaccinated susceptible who did not gained immunity ($S_1$, light orange) vaccinated susceptible who already lost immunity ($S_2$, dark orange), vaccinated immunized ($V$, green). Moreover, by $I_\Sigma$ (red) and $R_\Sigma$ (blue) we denote all infected and all recovered (independently of vaccination result), respectively. **c** Time evolution of the *instantaneous reproduction number* $\mathcal{R}^*$ (y axis) depending on the number of days counted from the start of the vaccination program (x axis), in five different scenarios describing the epidemic evolution: overcritical ($+$, red, $f = 0.77$ and $f_v = 0.38$), subcritical ($-$, blue, $f = 0.92$ and $f_v = 0.71$), initially and eventually overcritical ($+ - +$, orange, the same restrictions as in **b**: $f = 0.77$ and $f_v = 0.55$), eventually overcritical ($-+$, pink, $f = 0.92$ and $f_v = 0.38$), and eventually subcritical ($+-$, cyan, with $f = 0.77$ and $f_v = 0.71$). As controls, two additional scenarios of the epidemic evolution are presented, corresponding to no implementation of VPs and no changes in behavior due to vaccination: subcritical (another example of - scenario, green) with $f = f_v = 0.92$ and eventually subcritical (another example of $+-$ scenario, yellow) with $f = f_v = 0.77$, both plotted with dot-dashed line.

people in $S_1$ are perceived as immunized, but in fact are susceptible. $S_1$ is populated from $S_N$ with rate $(1 - a)v$. The vaccinated from the $V$ group move to the $S_2$ group of susceptibles with immunity waning rate $\omega$. The individuals from the $S_1$ group move to $S_2$ with the same rate $\omega$ to ensure that the ineffectively vaccinated are revaccinated with the same speed as the ones for which the vaccine was effective. The $S_2$ group is the group of vaccinated, but no longer immune, and thus, susceptible individuals. In contrast to $S_1$, we consider that the $S_2$ group is subject to revaccination. Consequently, a fraction of size $a$ of the population from $S_2$ populates $V$ with rate $av_r$ and a fraction of size $(1 - a)$ populates $S_1$ with rate $(1 - a)v_r$. Across the manuscript, we assume $v_r = v$, but the model is general and different values can be considered.

Some of the susceptibles in $S_1$ (or, similarly, $S_2$) may not get revaccinated fast enough and may become infected and populate $I_1$ (or, $I_2$). Then, as in the classical SIRS model, the $I_1$ (or $I_2$) population recovers and populates group $R_V$ with rate $\gamma$. We consider that the recovered in $R_V$ may also lose the immunity, and become susceptible again and move to $S_2$ with rate $\kappa$. The remaining susceptible subgroups (the $S_N$ and $S_D$) may undergo the same classical dynamics, i.e., become infected, recover, and either become susceptible again or, in case of the recovered in the $R_N$ subgroup, become vaccinated with rate $v$.

The following parameters are used to describe population dynamics in the model:

| | |
|---|---|
| $f_v, f$ | restrictions level (for VP holders and others) |
| $\beta_0$ | basic transmission rate |
| $\beta_v, \beta$ | transmission rate (for VP holders and others) |
| $\gamma$ | recovery rate |
| $\kappa$ | natural immunity waning rate |
| $a$ | vaccination effectiveness |
| $v$ | vaccination rate |
| $v_r$ | revaccination rate |
| $\omega$ | vaccine−induced immunity waning rate |
| $d$ | fraction of population that will never get vaccinated |

where the transmission rates are expressed int terms of the basic transmission rate and the restriction level parameters:

$$\beta = \beta_0(1 - f),$$
$$\beta_V = \beta_0(1 - f_V)$$

Finally, the following set of ordinary differential equations (ODEs) defines the dynamics:

$$\frac{d}{dt}S_D = -(\beta I + \beta I_V)S_D + \kappa R_D,$$
$$\frac{d}{dt}S_N = -(\beta I + \beta I_V)S_N - \upsilon S_N + \kappa R_N,$$
$$\frac{d}{dt}S_1 = \upsilon_r(1-a)S_2 + \upsilon(1-a)S_N - \omega S_1$$
$$\qquad - (\beta I + \beta_v I_V)S_1,$$
$$\frac{d}{dt}S_2 = -\upsilon_r S_2 + \omega V + \omega S_1 - (\beta I + \beta_v I_V)S_2 + \kappa R_V,$$
$$\frac{d}{dt}V = \upsilon a S_N + \upsilon_r a S_2 - \omega V + \upsilon_r R_V + \upsilon R_N,$$
$$\frac{d}{dt}I_D = (\beta I + \beta I_V)S_D - \gamma I_D, \qquad (1)$$
$$\frac{d}{dt}I_N = (\beta I + \beta I_V)S_N - \gamma I_N,$$
$$\frac{d}{dt}I_1 = (\beta I + \beta_v I_V)S_1 - \gamma I_1,$$
$$\frac{d}{dt}I_2 = (\beta I + \beta_v I_V)S_2 - \gamma I_2,$$
$$\frac{d}{dt}R_V = \gamma I_V - \kappa R_V - \upsilon_r R_V,$$
$$\frac{d}{dt}R_D = \gamma I_D - \kappa R_D,$$
$$\frac{d}{dt}R_N = \gamma I_N - \kappa R_N - \upsilon R_N,$$

where also the following relations hold

$$\begin{aligned} S_V &= S_1 + S_2, \\ I_V &= I_1 + I_2, \\ S &= S_D + S_N, \\ I &= I_D + I_N, \\ R &= R_N + R_D, \end{aligned}$$

with the constraint $S, S_V, I, I_V, R, R_V \geq 0$. Finally, to consider the subpopulation dynamics in terms of fractions of the entire subpopulation, we set

$$S + S_V + I + I_V + R + R_V + V = 1 \qquad (2)$$

and denote $d$ to be the fraction of the never-vaccinated population

$$d = S_D + I_D + R_D.$$

The endemic state of the VAP-SIRS model is computed in the Supplementary Note 1.

*Modeling restrictions.* We assume that the VP holders consist of the following subpopulations of vaccinated at least once: $V, S_V, I_V, R_V$. Recall that the net effect of all non-pharmaceutical interventions is modeled using parameters $f_v$ and $f$, called restrictions throughout the text. The parameter $f_v$ amounts to the level of restriction of contacts, and thus the ability to infect, within the group of VP holders. The parameter $f$ satisfies $f \geq f_v$ and corresponds to restriction of contacts within the rest of the population, as well as between the VP holders and the rest of the population.

The restriction level $f_v$ for the VP holders is introduced in the model as a modulator of the transmission rate $\beta_v$. Specifically, we assume that $\beta_v = \beta_0(1-f_v)$, where $\beta_0$ is the transmission rate of the SARS-CoV-2 virus without restrictions. We assume $f_v$ ranges from 0 to 1, where $f_v = 0$ corresponds to no restrictions enforced on the VP holders, and $f_v = 1$ corresponding to full restrictions. Given that for $f_v = 0$ the reproduction number $R_{\max} = \beta_0/\gamma$, and that the recovery rate $\gamma = 1/6$, we obtain the no-restriction transmission rate $\beta_0 = R_{\max}/6$. Thus, for the Delta variant, with $R_{\max} = 6$, $\beta_0 = 1$. Similarly, the transmission rate parameter $\beta = \beta_0(1-f)$ describes the transmission rate within the rest of the population and between VP holders and the rest, given the restrictions $f$.

*Proportional versus preferential types of social mixing.* The above described model equations are based on the assumption that the social mixing between social groups in the population is proportional to the group sizes (the mass action principle). Instead, preferential mixing can be assumed, where the VP holders are more likely to contact other VP holders, since they have lower restrictions[45]. This preferential bias is proportional to the difference between the restrictions $f$ and $f_v$. Preferential mixing is a common, socio-psychological motivated mixing scheme alternative to proportional mixing. In this scheme the group interaction is still proportional, but biased by the relative degree of freedom given to the passport holders. Preferential mixing as a modulation of proportional mixing was previously studied in the context of infectious diseases by Glasser et al.[46]. To incorporate the preferential mixing effect in the ODE model (Eq. (1)) we rescale the interaction terms according to the following rules:

$$S_V I_V \quad \rightarrow \quad \frac{\beta_v}{\beta(S+I+R)+\beta_v(1-(S+I+R))} S_V I_V$$
$$S I_V \quad \rightarrow \quad \frac{\beta}{\beta(S+I+R)+\beta_v(1-(S+I+R))} S I_V,$$

where $S + I + R$ is the non-immune population.

**Numerical integration and parameter values**. For simulations, we solve the model numerically by means of joint Adams' and BDF methods, as implemented in the R package deSolve, lsoda method of the ode function[47]. The method monitors data in order to select between non-stiff (Adams') and stiff (BDF) methods. It uses the non-stiff method initially[48].

To generate the data presented in Fig. 1b, we use the reference setup of parameters for the Delta variant: $\beta_0 = 1$, $f = 0.77$ (and thus $\beta = 0.23$), $f_v = 0.55$ (and thus $\beta_v = 0.45$), $\gamma = 1/6$, $\kappa = 1/500$, $a = 0.79$, $\upsilon = \upsilon_r = 1/250$, $\omega = 1/500$, $d = 0.12$, with initial conditions $I = 10^{-6}$, $I_D = d \cdot I = 10^{-7}$; $I_N = (1-d) \cdot I = 0.9 \cdot 10^{-6}$, $R = 0$, $V = 0$. Given $I(t)$ resulting from the solution of the model's ODE system, to present the final results as easier interpretable cases per million rather than fractions, we re-scale the results by 1M. Additionally, we compute a proxy for the daily incidence number of new cases from the following relation between $I(t)$ and $I^*(t)$:

$$I(t) = \int_0^t e^{-\gamma(t-\tau)} I^*(\tau) d\tau$$
$$= \int_{t-1}^t e^{-\gamma(t-\tau)} I^*(\tau) d\tau + e^{-\gamma} \int_0^{t-1} e^{-\gamma(t-1-\tau)} I^*(\tau) d\tau$$
$$\simeq \frac{1}{\gamma} I^*(t)(1 - e^{-\gamma}) + e^{-\gamma} I(t-1).$$

Thus, the $I^*(t)$ is computed as

$$I^*(t) \simeq \frac{\gamma}{1 - e^{-\gamma}} (I(t) - e^{-\gamma} I(t-1)).$$

We proceed similarly to obtain daily incidence numbers $I_1^*$, $I_2^*$ and for the sum of all infected, and again to make it interpretable in the figures we re-scale it by 1M.

**Stability analysis**. The vaccination dynamics can be solved explicitly in the absence of infections. Fixing $I = I_V = R = R_V = 0$, and assuming $\upsilon = \upsilon_r$, we obtain

$$S(t) = d + (1-d)e^{-\upsilon t},$$
$$V(t) = (1-d)\frac{\upsilon a}{\upsilon a + \omega}\left(1 - e^{-(\upsilon a + \omega)t}\right),$$
$$S_V(t) = 1 - S - V.$$

For convenience, where it is not needed, we drop the time argument.

Taking an adiabatic approach we linearize the infection dynamics for small $I$, $I_V$ and $R$ under the assumption of slowly

varying $S$, $S_V$ and $V$. In that case, the infection dynamics decouples from the vaccination dynamics and the Jacobian submatrix $J_{sub}$ for the equations for $I$ and $I_V$ is given by:

$$J_{sub} = \begin{pmatrix} \beta S - \gamma & \beta S \\ \beta S_V & \beta_V S_V - \gamma \end{pmatrix}.$$

Given the Jacobian submatrix, we can approximate the dynamics in a small neighborhood of the $I = I_V = 0$ state as

$$\begin{pmatrix} \frac{d}{dt} I \\ \frac{d}{dt} I_V \end{pmatrix} = \begin{pmatrix} \beta S - \gamma & \beta S \\ \beta S_V & \beta_v S_V - \gamma \end{pmatrix} \cdot \begin{pmatrix} I \\ I_V \end{pmatrix}. \qquad (3)$$

*The instantaneous reproduction number $\mathcal{R}^*$ and the instantaneous doubling time D.* Since both the eigenvalues $\lambda_{max}$ and $\lambda_2 \le \lambda_{max}$ of $J_{sub}$ are real, the solution to Eq. (3) providing the dynamics of infection numbers of the vaccinated and the rest of the population in time can be written in the following form

$$\begin{pmatrix} I(t) \\ I_V(t) \end{pmatrix} = c_1 w_1 e^{\lambda_{max} t} + c_2 w_2 e^{\lambda_2 t}$$
$$= e^{\lambda_{max} t}(c_1 w_1 + c_2 w_2 e^{(\lambda_2 - \lambda_{max})t}),$$

where $w_1$ and $w_2$ are the respective eigenvectors, and $c_1$ and $c_2$ are constants depending on the initial conditions.

Since we have $\lambda_2 - \lambda_{max} \le 0$, we can approximate the time evolution of infection numbers by

$$\begin{pmatrix} I(t) \\ I_V(t) \end{pmatrix} \approx c_1 w_1 e^{\lambda_{max} t}. \qquad (4)$$

The largest eigenvalue of $J_{sub}$ is given by

$$\lambda_{max} = \frac{1}{2} S\beta - \gamma + \frac{1}{2} S_V \beta_v$$
$$+ \frac{1}{2} \sqrt{S^2 \beta^2 + S_V^2 \beta_v^2 - 2SS_V \beta\beta_v + 4SS_V \beta^2},$$

whereby it is convenient to express $\lambda_{max}$ as a function of $R_1 = \frac{\beta}{\gamma}$ and $R_2 = \frac{\beta_v}{\gamma}$. We then obtain

$$\lambda_{max} = \frac{\gamma}{2} \sqrt{(R_1 S - R_2 S_V)^2 + 4SS_V R_1^2}$$
$$+ \frac{\gamma}{2}(R_1 S + R_2 S_V) - \gamma. \qquad (5)$$

We now describe the relation of the analyzed system with the corresponding branching process, which motivates the notion of the instantaneous reproduction number and the derivation of the doubling time. It also allows a straightforward generalization to more complex systems of equations than the one considered here. Given the population fractions $S(t)$ and $S_V(t)$ at a given time instant $t$, the linearized dynamics of infections given by Eq. (3) has a corresponding two-type Galton-Watson branching process, which is a microscopic description of the dynamics. The two types of the process correspond to the $I$ and $I_V$ groups. The type $I$ individuals generate $Pois(R_1 S)$ offsprings of type $I$ and $Pois(R_1 S_V)$ offsprings of type $I_V$. The type $I_V$ individuals generate $Pois(R_1 S)$ offsprings of type $I$ and $Pois(R_2 S_V)$ offsprings of type $I_V$. The linearized dynamics (3) can then be understood as a mean field limit of the microdynamics described by such a branching process. Moreover, the spectral norm

$$\mathcal{R}^* = \frac{1}{2}(R_1 S + R_2 S_V) + \frac{1}{2} \sqrt{4R_1^2 SS_V + (R_1 S - R_2 S_V)^2} \qquad (6)$$

of the transition matrix

$$\begin{pmatrix} R_1 S & R_1 S_V \\ R_1 S & R_2 S_V \end{pmatrix}$$

of the branching process can be interpreted as the reproduction number of the branching process, since the expected number of infected in generation $n$ grows like $const \cdot (\mathcal{R}^*)^n$[49]. We refer to $\mathcal{R}^*$ as the instantaneous reproduction number. The term instantaneous comes from the fact that we are considering the linearized adiabatic dynamics in a small neighborhood of the $I = I_V = 0$ (ref Eq. (3)).

The above discrete branching process can be extended to a continuous time branching process by assuming a probability distribution on the generation time, denoted $\varphi(\gamma)$. The growth of the continuous time branching process $const \cdot e^{\alpha t}$ is characterized by its Malthusian growth parameter, denoted $\alpha$. The relation between the instantaneous reproduction number $\mathcal{R}^*$, the distribution $\varphi(\tau)$ and the Malthusian parameter $\alpha$ for such a branching process is given by

$$\mathcal{R}^* \cdot \mathcal{L}_\varphi(\alpha) = 1$$

where $\mathcal{L}_\varphi(\alpha)$ is the Laplace transform $\int_0^\infty e^{-\alpha\tau} \varphi(\tau) d\tau$ of the distribution $\varphi$[49]. Since the setting of ODE model (1) implies exponential distribution of the generation time, i.e, $\varphi(\gamma) = Exp(\gamma)$, the following relation holds: $\alpha = \gamma(\mathcal{R}^* - 1)$.

By Eq. (4), the Malthusian parameter $\alpha$ for our dynamics is given by the largest eigenvalue $\lambda_{max}$. Hence we obtain the relation between the instantaneous reproduction $\mathcal{R}^*$ and the $\lambda_{max}$ as $\lambda_{max} = \gamma(\mathcal{R}^* - 1)$. Note that since both $S$ and $S_V$ are functions of time, so are $\lambda_{max}$ and $\mathcal{R}^*$.

It is noteworthy that in the above equations, all $R_1$, $R_2$, $R_1 S$ and $R_2 S_V$, and $\mathcal{R}^*$ should be seen as reproduction numbers, but of a different nature[50]. $R_1$ and $R_2$ are reproduction numbers taking into account the restrictions $f$ and $f_v$, respectively. The $R_1 S$ and $R_2 S_V$ are also group specific, but in addition incorporate the respective group sizes. Finally, $\mathcal{R}^*$ combines all these factors together.

Having this and Eq. (4), we define the instantaneous doubling time at time, denoted $tD(t)$, as the solution $D$ of $e^{\gamma(\mathcal{R}^*(t)-1) \cdot D} = 2$. Such obtained doubling times are featured in Supplementary Fig. S1.

Only a small change is needed in the derivation to extend to more complex systems than the considered SIRS model. For example, in a dynamics with exponential $Exp(u)$-distributed additional incubation time $u$ and exponential $Exp(c)$-distributed duration of the infectious period $c$ (a so called SEIRS model), and a given $\mathcal{R}^*$, we would have for the Malthusian growth parameter $\alpha$ the relation $(\alpha + c)(\alpha + u) = uc\mathcal{R}^*$, from which one can easily compute the corresponding doubling time.

*The times of transitions between subcritical and overcritical epidemics.* The analysis of the linearized dynamics around $I = I_V = 0$ allows us to determine transitions between subcritical and overcritical epidemics. Such transitions occur at the time instants $t$ at which $\lambda_{max}(t) = 0$, or, equivalently, at $\mathcal{R}^*(t) = 1$. We thus find that for given values of $S(t)$ and $S_V(t)$ the critical times $t$ for transitions between subcritical and overcritical epidemics are the roots of the equation

$$\lambda_{max}(t) = 0.$$

The obtained critical threshold times are plotted in the lower triangles of the panels in Fig. 2 and Supplementary Fig. S3 in the main text. In the case of proportional mixing the above equation is equivalent to:

$$(R_1 S(t) - 1)(R_2 S_V(t) - 1) = R_1^2 S(t) S_V(t).$$

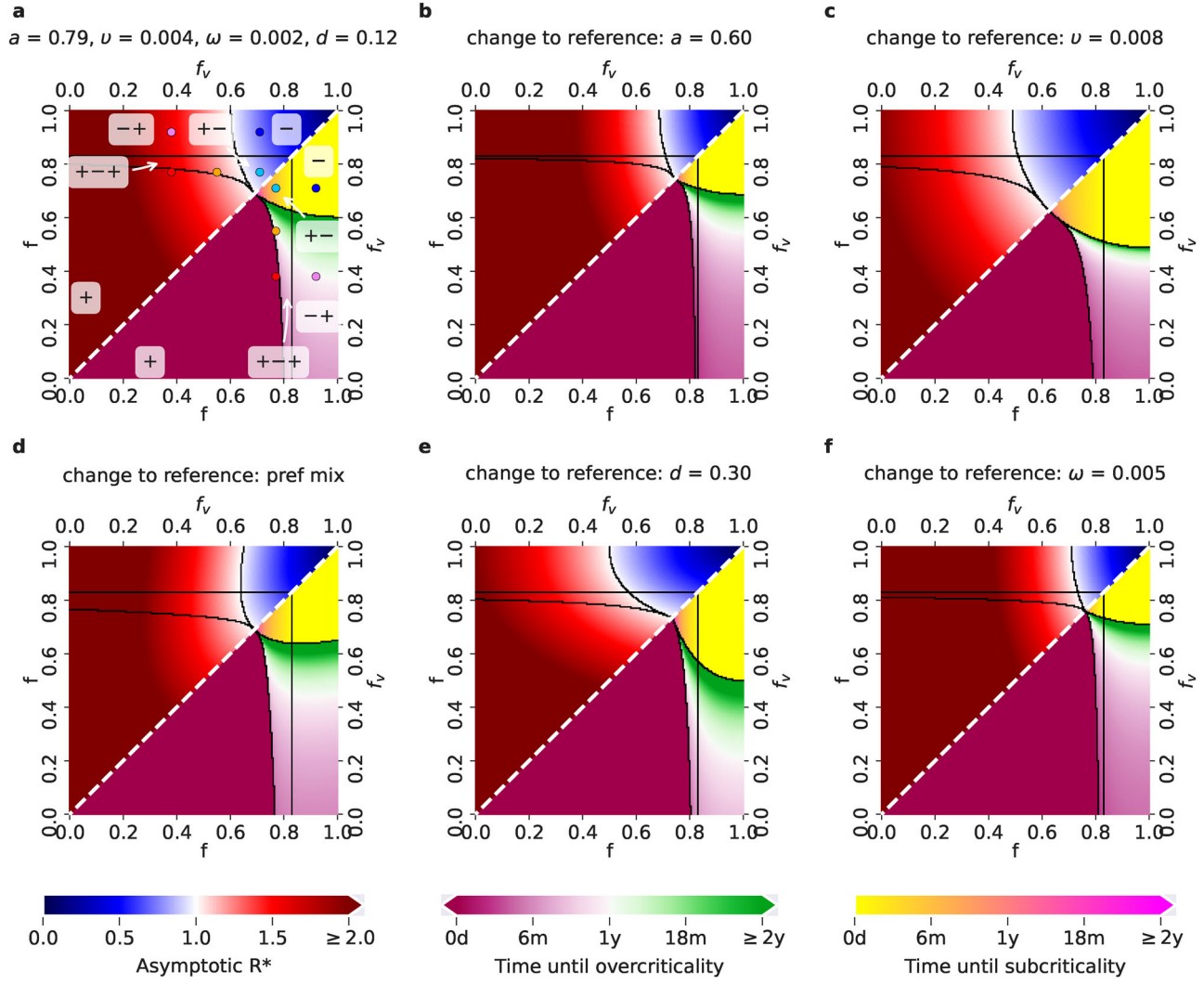

**Fig. 2 Possible COVID-19 epidemic dynamics for different parameter setups for the Delta variant.** The relevant $f - f_v$ parameter space, where $f_v \leq f$, can be divided into five regions (delimited by black borders), each associated with a different behavior of the epidemics. On the diagonal (white dashed line), $f = f_v$, i.e., the restrictions for VP holders and for the rest of the population are the same - corresponding to the situation when VPs are not introduced at all. Lower triangles show the time until the last critical threshold: different color scales correspond to the time until the switch either from a subcritical to an overcritical epidemic (time until overcriticality, violet-green scale), or from an overcritical to a subcritical epidemic (time until subcriticality, yellow-pink scale). Upper triangles show the asymptotic $\mathcal{R}^*$, as a function of the values of $f$ and $f_v$ (blue-red scale, with blue associated with $\mathcal{R}^*<1$ and red associated with $\mathcal{R}^*>1$). **a** Reference setup, with $a = 0.79$ (corresponding to the effectiveness of the Comirnaty vaccine on the Delta variant), $\upsilon = \upsilon_r = 0.004$, $\omega = 0.002$, $d = 0.12$ (fraction of never-vaccinated in the United Kingdom) and proportional mixing. The choices of $(f, f_v)$ corresponding to the five scenarios exemplified in Fig. 1c are denoted by points of the same color. **b** Setup with a decreased vaccine effectiveness: $a = 0.6$ (corresponding to the effectiveness of the Vaxzevria vaccine on the Delta variant). **c** Setup with an increased vaccination rate: $\upsilon = \upsilon_r = 0.008$. **d** Setup with preferential (instead of proportional) mixing. **e** Setup with an increased fraction of people who will not get vaccinated: $d = 0.3$ (fraction of never-vaccinated in France). **f** Setup with an increased waning rate: $\omega = 1/200$.

*Asymptotic structure of the population and minimum common restrictions required to avoid epidemic resurgence.* The asymptotic structure of the population in terms of the sizes of the sub-populations $V$, $S_V$ and $S_D$ can be easily obtained by setting $I = I_V = R = R_V = 0$ and computing the stable stationary solution for $V^{as}$, $S^{as}$ and $S_V^{as}$ of our ODE system (1):

$$S^{as} = d$$
$$S_V^{as} = (1-d)(1-\eta)$$
$$V^{as} = (1-d)\eta$$
$$S^{as} + S_V^{as} = 1 - V^{as},$$

where

$$\eta = \frac{a}{1 + \omega/\upsilon_r}$$

can be seen as the actual immunization rate in the population, and is expressed as a function of vaccine effectiveness $a$ and the ratio of the immunity waning rate $\omega$ and the revaccination rate $\upsilon_r$. The obtained values correspond to the structure in the limit $t \to \infty$ and represent the structure to which the population converges in the long term.

Having this, we obtain the asymptotic instantaneous reproduction number $\mathcal{R}^*$ by inserting the asymptotic values $S^{as}$ and $S_V^{as}$ into Eq. (6). These values are plotted in the upper triangles in the panels of Fig. 2 and Supplementary Fig. S3 in the main text.

Finally, we solve for such minimum common restrictions $f = f_v = f_{min}$, which will result in instantaneous reproduction number $\mathcal{R}^* = 1$ for the different vaccine effectiveness and vaccination rate setups. Hence $f_{min}$ is found from $R_{max}(1 - f_{min}) = \frac{1}{1-V}$ as

$$f_{min} = \max\left(0, 1 - \frac{1}{R_{max}(1 - V^{as})}\right).$$

**Reporting summary**. Further information on research design is available in the Nature Research Reporting Summary linked to this article.

## Results

**The VAP-SIRS model of the impact of COVID-19 VPs**. The proposed VAP-SIRS model extends the classical SIRS model[44] (red arrows in Fig. 1a) with additional states and parameters that describe the dynamics of vaccination rollout in a population (green arrows in Fig. 1a). To this end, we consider the following subpopulations: (i) initially susceptible $S_N$, who, if successfully vaccinated, populate the immune group $V$, with rate $av$, where $v$ is the vaccination rate and $a$ is the vaccination effectiveness, (ii) susceptible who were vaccinated but did not gain immunity ($S_1$), (iii) vaccinated, whose immunity waned with rate $\omega$ and who became susceptible again ($S_2$), (iv) susceptible, who are not and will never get vaccinated ($S_D$). The $S_D$ compartment contains people who for health reasons cannot receive current types of vaccines, as well as individuals who do not get vaccinated because of hesitancy, beliefs or other individual reasons. The fraction of the population that will never be vaccinated is denoted by $d$. Additionally, revaccination of $S_2$ populates $V$ with rate $av_r$. All recovered, unless in the recovered compartment $R_D$, are also subject to vaccination. Before the recovered in the $R_V$ lose immunity, they might be revaccinated, and, thus, populate the $V$ group with rate $v_r$ (similarly, $R_N$ are vaccinated with rate $v$). In this case, vaccination effectiveness is fixed to 1, which is substantiated on the basis of the fact that vaccination combined with a previous infection should confer a much stronger protection than only vaccination of a susceptible individual. Across the manuscript, we assume the revaccination and vaccination rates are equal, $v_r = v$. Additional sub-compartments are used to model hospitalizations (see the Supplementary Note 1).

The presented model analysis is performed for carefully selected parameter setups. We consider two different vaccination rates $a$, 0.004 and 0.008 doses per person daily, chosen on the basis of the current rates observed in Europe[51,52]. As vaccine effectivenesses for the Delta variant, we consider 0.6 and 0.79, which were reported as the effectivenesses of the most widely used vaccines: Vaxzevria (AstraZeneca) and Comirnaty (BioN-Tech/Pfizer) respectively for this variant[20]. For the Alpha variant, the effectivenesses of the same vaccines for that variant are considered instead, namely 0.79 (Vaxzevria) and 0.92 (Cominarty)[20]. We consider realistic fractions $d$ of never-vaccinated equal to 0.12 (optimistic), and 0.3 (pessimistic), reported for the United Kingdom and France, respectively [https://ourworldindata.org/, as of June 15th, 2021]. Furthermore, two post-vaccination immunity waning rates $\omega$ are considered corresponding to optimistic (500 days; $\omega = 1/500$) and pessimistic (200 days; $\omega = 1/200$) average duration of immunity against infection, reflecting emerging data on large individual variation of immunity waning and other key factors influencing this process[15,18,24–26]. There remains uncertainty regarding the waning time for natural immunity, and whether it varies between the different SARS-CoV-2 variants, but early evidence indicates it lasts at least 180 days[15,53,54]. Hence, we consider an optimistic scenario of natural immunity lasting on average similarly long as the optimistic immunity gained via vaccination: 500 days (corresponding to natural immunity waning rate $\kappa = 0.002$). Based on the current studies, we fix the generation time to 6 days ($\gamma = 1/6$)[10,55].

We assume that VP holders are all those who completed at least one complete vaccination cycle, i.e., one dose or two doses depending on the vaccine used (Fig. 1), which is also the basis on which the EU DCC is issued. The restriction level (ranging from 0 to 1) is introduced as a modulator of the SARS-CoV-2 reproduction number. Here, we consider that without any restrictions, the basic reproduction number for the B.1.617.2 variant (Delta) is equal to 6 (an optimistic estimate based on refs. [56,57]), while for the B.1.1.7 variant (Alpha) an optimistic estimate is equal to 4[10,55]. Two levels of restrictions are considered: restrictions $f_v$ for contacts among VP holders, as well as restrictions $f$ for contacts of the VP holders with the rest of the population and for contacts within the rest of the population. The impact of VPs is studied assuming that $f_v < f$: a VP holder has more freedom of contact with other VP holders, or is generally subject to fewer restrictions on the VP holders than the rest of the population. Importantly, in general $f$ and $f_v$ should be interpreted as the net effect of all combined factors that reduce the reproduction number of the virus within the respective groups: all applied non-pharmaceutical interventions, including testing and isolation, together with the resulting changes in behavior. The situation where no VPs are implemented, hence the vaccinated have the same restrictions as the rest of the population, and there are no changes of behavior due to vaccination, is modeled by fixing $f_v = f$. Finally, to analyze the impact of social behavior, we consider two types of mixing between subpopulations: proportional (typical for SIR models) and preferential, where the VP holders prefer contacts with other VP holders. See Methods for a detailed model description.

**VAP-SIRS predicts a possible infection resurgence despite vaccinations**. VAP-SIRS predicts unfavorable epidemic dynamics for a wide range of parameters, both for the Delta and the Alpha variants. As an example consider the Delta variant, and vaccine effectiveness $a = 0.79$ (the effectiveness of the Comirnaty vaccine against the Delta variant), (re-)vaccination rates $v = v_r = 0.004$, low never-vaccinated fraction $d = 0.12$ (reported for the United Kingdom), low immunity waning rate $\omega = 0.002$, low natural immunity rate loss $\kappa = 0.002$, and proportional mixing. This set of parameters corresponds to a seemingly safe setup, which we will call the reference setup. The impact of various parameter changes with respect to this reference will be considered below. For such a setup, consider medium-high restrictions level $f = 0.77$ for contacts of the VP holders with the rest of the population and within the rest of the population, along with a restrictions reduction for the VP holders compared to the rest of the population by around 30%, resulting in medium restrictions $f_v = 0.55$ for VP holders. For these parameters, the model predicts a small wave of infections shortly after the vaccination program starts, followed by a large wave later (color curves Fig. 1b). This behavior is explained by the population structure (Muller plot, Fig. 1b) and can only happen due to the different levels of restrictions for the VP holders and the rest of the population. In this scenario, the first wave is driven by the unvaccinated susceptibles ($S_N$) and suppressed by ongoing vaccination, as expected. Interestingly, the second, larger wave is driven by the $S_V$ group. The $S_V$ group is composed of the number of individuals for whom the vaccine was ineffective ($S_1$) and those vaccinated who lose their immunity and are not yet revaccinated ($S_2$).

**Stability analysis identifies potential scenarios for the COVID-19 epidemic depending on the restrictions imposed on VP**

**holders and the rest of the population**. To assess the epidemic evolution in different scenarios, we analyse stability by linearizing the model equations with $I = R = 0$ and introduce the *instantaneous reproduction number* $\mathcal{R}^*$ (see Methods). $\mathcal{R}^*(t)$ is the reproduction number that would be observed at time $t$, given the restrictions $\mathbf{f} = (f, f_v)$ and the composition of the population (the population fractions of the groups: susceptible with VPs $S_V$, susceptible without VPs $S$, and the immune group $V$), where the number of infected is very small. For $\mathcal{R}^*(t) > 1$, switching to $\mathbf{f}$ at time $t$ results in an *overcritical* epidemic evolution, with an initially exponential growth of infections; for $\mathcal{R}^*(t) < 1$, switching to $\mathbf{f}$ at time $t$ results in a *subcritical* epidemic evolution, where the number of active cases decreases to zero. The $\mathcal{R}^*$ is more informative of epidemic thresholds than the standard effective reproduction number, as it does not depend on the actual number of infected and recovered.

Assuming the reference setup for the Delta variant, we consider five choices of restriction combinations (prototypical for five regions of the parameter space, see Fig. 2), leading to different time profiles of $\mathcal{R}^*$ (Fig. 1c). As control setups, we introduce two settings that represent policies in a given population without the implementation of the VPs, one with a common high restriction level $f = f_v = 0.92$, which keeps the epidemic subcritical (scenario denoted −, green dot-dashed line in Fig. 1c), and one with a common medium-high restriction level $f = f_v = 0.77$, which results in a time evolution of $\mathcal{R}^*$ from overcritical to subcritical (denoted +−, yellow dot-dashed line in Fig. 1c). In such settings, in case VPs are introduced, VP holders can gain low (<20%), medium (20-50%) or high (> 50%) restriction reduction with respect to the restrictions for the rest of the population. Granting high restriction reductions to VP holders, both with mid-high and with high restrictions enforced for the rest of the population, eventually leads to an overcritical epidemic (red and pink curve in Fig. 1c: the red curve shows a persistent overcritical epidemic, a scenario denoted +, while the pink curve shows an epidemic that is initially subcritical and then becomes overcritical, a scenario denoted −+). Medium restriction reductions for VP holders, along with high restrictions for the rest of the population, yield a subcritical epidemic evolution (another example of scenario −, blue curve in Fig. 1c). When mid-high restrictions are enforced for the rest of the population, a medium restriction reduction for VP holders leads to an epidemic that is initially overcritical, then becomes subcritical and after a few months switches to overcritical again, starting a new wave of infections (orange curve in Fig. 1c; denoted +−+, this is also the scenario shown in the simulation in Fig 1b). Finally, always with mid-high restrictions enforced for the rest of the population, a low restriction reduction for VP holders leads to an epidemic that is initially overcritical and then switches to subcritical (another example of scenario +−, cyan curve in Fig 1c).

In each scenario we computed the time evolution of the *instantaneous doubling time* $D$, capturing how fast the infections grow. For a given $\mathbf{f}$, $D(t)$ is the doubling time that would be observed for the growth of a small initial number of infections at time $t$, with enforced restrictions $\mathbf{f}$. Very short doubling times, below 30 days, can be observed in three scenarios that are (eventually) overcritical: see the red, orange and pink curves in the Supplementary Fig. S1.

**Flexible measures are required to avoid epidemic resurgence depending on parameter setups**. The relevant $f − f_v$ parameter space, where $f_v \leq f$, can be divided into five regions, where the epidemic dynamics follows the distinct patterns exemplified in Fig. 1c. Fig. 2 shows the impact of changing specific single parameter values on the expected scenarios and on times to

critical events, tracking time up to two years. The area occupied by each region changes depending on the parameter setups. For example, in the reference setup for the Delta variant ($a = 0.79$ - the Comirnaty effectiveness on the Delta variant, $v = v_r = 0.004$, $d = 0.12$ - the fraction of never vaccinated in the United Kingdom, $\omega = 1/500$, $\kappa = 1/500$, and proportional mixing) in Fig. 2a, the overcritical region (denoted +, with $\mathcal{R}^*$ always above 1) occupies the lower left corner. This region is enlarged in the case of a lower vaccine effectiveness ($a = 0.6$ - the effectiveness of Vaxzevira on the Delta variant, Fig. 2b), and higher waning rate (Fig. 2f). In contrast, it shrinks with a higher vaccination rate (Fig. 2c), indicating that there is a concrete benefit from deploying efficient vaccination programs. The subcritical region (−, with $\mathcal{R}^*$ always smaller than 1) lies in the opposite corner of the $f − f_v$ space, for larger restriction values, and, for a fixed fraction of never-vaccinated $d$, tends to decrease for setups where the overcritical region increases. As expected, the switch to a larger fraction of never-vaccinated (to $d = 30\%$, corresponding to the reported fraction in France), increases the overcritical (+) region (Fig. 2e). But, at the same time, the larger fraction of never-vaccinated increases also the subcritical (−) region. This is due to the fact that the never-vaccinated are assumed to follow stricter restrictions, compared to VP holders, and therefore their larger fraction can constrain the emergence of the later waves, characteristic of the regions +−+ and −+. Still, a strategy relying on this effect might be difficult to implement due to the large + region and can lead to undesirable outcomes in practice.

Inside each of the three regions associated with the +−+, −+, +− scenarios in Fig. 1c, the specific parameter settings differ by the time to the critical threshold of interest for that region (the last observed switch between subcritical and overcritical epidemic, which for the +−+ region, for example, is the second critical threshold; see Methods for the computation of the times to critical thresholds). For the reference setup (Fig. 2a) and the +−+ region, the critical threshold is reached after a minimum ~4 months. Decreasing the vaccine effectiveness from Comirnaty's to Vaxzevira's (Fig. 2b), as well as increasing the waning rate (Fig. 2f), leads to overcriticality sooner, after a minimum of ~ 2 and ~ 3 months respectively, for low $f_v$ values. Increasing vaccination rate (Fig. 2c) shrinks the +−+ region. The comparison between proportional and preferential mixing shows the impact of more intense interactions of the VP holders inside of their own group, and less intense contacts of the VP holders with the rest of the population. With preferential mixing (Fig. 2d), the +−+ region becomes larger and overcriticality is reached even sooner. This is due to the fact that preferential contacts among VP holders accelerate the emergence of the wave caused by infections of the VP holders. Seemingly counter-intuitively, increasing the number of never-vaccinated people (Fig. 2e) shrinks the +−+ region and delays the onset of overcriticality. This is due to the fact that the onset of overcriticality in the +−+ region depends not only on the intensity of contacts of the VP holders, but also on their fraction in the population; with a larger fraction of never-vaccinated, the fraction of VP holders in the population decreases.

The above analysis of the different regions predicts a possible switch to overcritical epidemic growth for a given parameter setup and, if there is a switch, it provides the time it happens, counting from the onset of the vaccination program. It does not, however, indicate how fast the overcritical growth will be. To inform about what growth rates can be eventually expected in the overcritical regime, we compute the asymptotic $\mathcal{R}^*$ (the $\mathcal{R}^*(t)$ for $t \to \infty$, see Methods) for all parameter setups and all combinations of restrictions in the relevant $f − f_v$ space. For a given restriction combination $\mathbf{f}$, the asymptotic $\mathcal{R}^*$ indicates how quickly the infections grow shortly after the restrictions are set to

**f** in the asymptotic state. For all considered parameter setups, except for the one with high (re-)vaccination rate, and for all except the +− and the − regions, large asymptotic $\mathcal{R}^*$ can be expected, which corresponds to short doubling times (Fig. 2). This analysis highlights the importance of avoiding the over-critical (+) region, as there the asymptotic $\mathcal{R}^*$ values can even exceed 2 when the restrictions are low.

Comparing Fig. 2 to Supplementary Fig. S2 shows how the Delta variant worsens all scenarios with respect to the Alpha variant: in all panels of Fig. 2, the Delta variant leads to a considerable expansion of the overcritical region, shrinking of the safe subcritical region, and to consistently larger values of asymptotic $\mathcal{R}^*$. This is due not only to a higher transmissibility of the Delta variant, but also due to the fact that the considered vaccines have lower effectiveness for this variant, as compared to the Alpha variant.

We further investigate how the expected scenarios, times to critical events (tracking time up to two years), and asymptotic $\mathcal{R}^*$ values are affected by changes of two parameters at once, compared to the reference setup, for the Delta (Supplementary Fig. S3) and the Alpha variant (Supplementary Fig. S4). The double parameter changes give insight into the possible synergistic and compensatory effects between individual para-meter changes. Compared to the effect of only decreasing the vaccine effectiveness from Comirnaty's to Vaxzevira's (Fig. 2b), the effect of jointly decreasing the vaccine effectiveness and increasing the vaccination rate (Supplementary Fig. S3a) indicates that a higher vaccination rate can compensate to some extent for the loss of effectiveness. Similarly, an increased vaccination rate can counteract increased immunity waning rate (Supplementary Fig. S3e). The combination of decreased effectiveness and increased immunity waning rate has the worst effect, as it largely increases the overcritical region (+), decreases the subcritical region (−) and accelerates the times to the overcriticality in all other regions (Supplementary Fig. S3c). Finally, combinations of an increased never-vaccinated fraction with other parameter changes show an interesting mix of effects. When both the never-vaccinated fraction and the vaccination rate increase, the overcritical (+) region decreases and the subcritical region increases, while the times to overcriticality in the +−+ and the −+ regions increase (Supplementary Fig. S3d). Similarly, there is a synergistic effect of the combination of the increased never-vaccinated fraction and the increased immunity waning rate (Supplementary Fig. S3f). For the Alpha variant, the effects of coupled parameter changes combine the same way as for the Delta variant, but once again it is apparent that, for all the parameter setups we considered, with the Alpha variant much less restrictions are required to avoid epidemic resurgence than with the Delta variant (Supplementary Fig. S4).

Taken together, these results indicate that, unless novel vaccines with higher effectiveness are invented and distributed, and unless much faster and wider vaccination programs are implemented, resulting in much more favorable parameter settings than the realistic ones analyzed here (including those considered optimistic), highly unfavorable infection dynamics are likely to emerge for the Delta variant, and less, but still, for the Alpha variant. The −+ and +−+ regions in Fig. 2 and Supplementary Fig. S3 can seem attractive as restriction policies, because they entail larger freedom for the VP holders; both these regions, however, eventually result in epidemic resurgence and either should be avoided or the time spent in these regions should be very carefully regulated. For example, if sufficient restrictions are enforced for the rest of the population, the VP holders may initially be granted additional freedoms (larger if the Alpha variant is dominant in the population, and much lower if the Delta variant is dominant), which corresponds to the −+ region.

**Table 1 Asymptotic level of immunization $V^{as}$ and minimum common restrictions $f_{min}$ for the Delta variant and different parameter setups.**

| Parameter setup | a | $v_r$ | d | $\omega$ | $V^{as}$ | $f_{min}$ |
|---|---|---|---|---|---|---|
| Ref. setup | 0.79 | 0.004 | 0.12 | 0.002 | 0.46 | 0.69 |
| Dec. a | **0.6** | 0.004 | 0.12 | 0.002 | 0.35 | 0.74 |
| Inc. $v_r$ | 0.79 | **0.008** | 0.12 | 0.002 | 0.56 | 0.62 |
| Inc. d | 0.79 | 0.004 | **0.3** | 0.002 | 0.37 | 0.74 |
| Inc. $\omega$ | 0.79 | 0.004 | 0.12 | **0.005** | 0.31 | 0.76 |
| Dec. a, inc. $v_r$ | **0.6** | **0.008** | 0.12 | 0.002 | 0.42 | 0.71 |
| Dec. a, inc. d | **0.6** | 0.004 | **0.3** | 0.002 | 0.28 | 0.77 |
| Dec. a, inc. $\omega$ | **0.6** | 0.004 | 0.12 | **0.005** | 0.23 | 0.78 |
| Inc. $v_r$, inc. d | 0.79 | **0.008** | **0.3** | 0.002 | 0.44 | 0.70 |
| Inc. $v_r$, inc. $\omega$ | 0.79 | **0.008** | 0.12 | **0.005** | 0.43 | 0.71 |
| Inc. d, inc. $\omega$ | 0.79 | 0.004 | **0.3** | **0.005** | 0.18 | 0.80 |

The studied parameters are: vaccine effectiveness a, revaccination rate $v_r$, fraction of never-vaccinated d, and waning immunity rate $\omega$. The first row concerns the reference setup; rows below are setups with the same parameters as in the reference setup, but with either one parameter changed (in bold; rows 2–5; same as in Figs. 2 and 3, apart from preferential mixing, as it is not relevant for common restrictions) or two parameters changed (in bold; rows 6–11). Dec. - Decreased, Inc. - increased.

In this way, an overcritical situation (region +) will be avoided. However, to prevent the epidemic from becoming overcritical after an initial decline in case numbers, restrictions on VP holders need to be timely increased and adapted, to avoid spending longer time in the −+ region than the time to overcriticality. Thus, moving out of the −+ region to the +− region with the right timing could be one of possible strategies. It may, however, be more practical to circumvent many changes of restriction policies over time and it may be fair for everyone to face the same restrictions. Safe common restrictions, however, corresponding to the parameters on the diagonal in the subcritical (−) region in Fig. 2 and Supplementary Figs. S2, S3 and S4, are relatively high, especially those required by the Delta variant, and may therefore cause unrest in the population.

**A minimum common restriction level can keep the epidemic subcritical in the long-term**. We compute the minimum com-mon restriction level $f_{min}$ for the whole population that would guarantee to avoid an overcritical epidemic in the long-term (for time approaching infinity, Methods):

$$f_{min} = \max\big(0, 1 - 1/(R_{max} \cdot (1 - V^{as}))\big),$$

where $V^{as}$ as is the asymptotic fraction of the immunized in the population

$$V^{as} = (1 - d)\frac{a}{1 + \omega/v_r}.$$

The resulting values differ depending on the setups of vaccine effectiveness a, revaccination rate $v_r$, the fraction of never-vaccinated population d and immunity waning rate $\omega$ (Table 1). The minimum common restrictions for the reference setup are equal to $f_{min} = 0.69$. Out of parameter setups with single change compared to the reference, doubled (re-)vaccination speed leads to the lowest possible common restriction level. Even for this most optimistic setup (high a = 0.79, high $v_r = 0.008$, low d = 0.12, low $\omega = 0.002$; Table 1 third row) we obtain $V^{as} = 0.6$, and $f_{min} = 0.62$. The level of 0.62 restrictions is around twice as high as the level 0.29 that would be required for the Alpha variant (Supplementary Table 1), and is a considerable reduction of freedom compared to before the pandemic. It is noteworthy that in the long term, to avoid infections rising, minimum common restrictions have to be increased to 0.74 with the larger fraction of never vaccinated d. Thus, a scenario with a large fraction of the

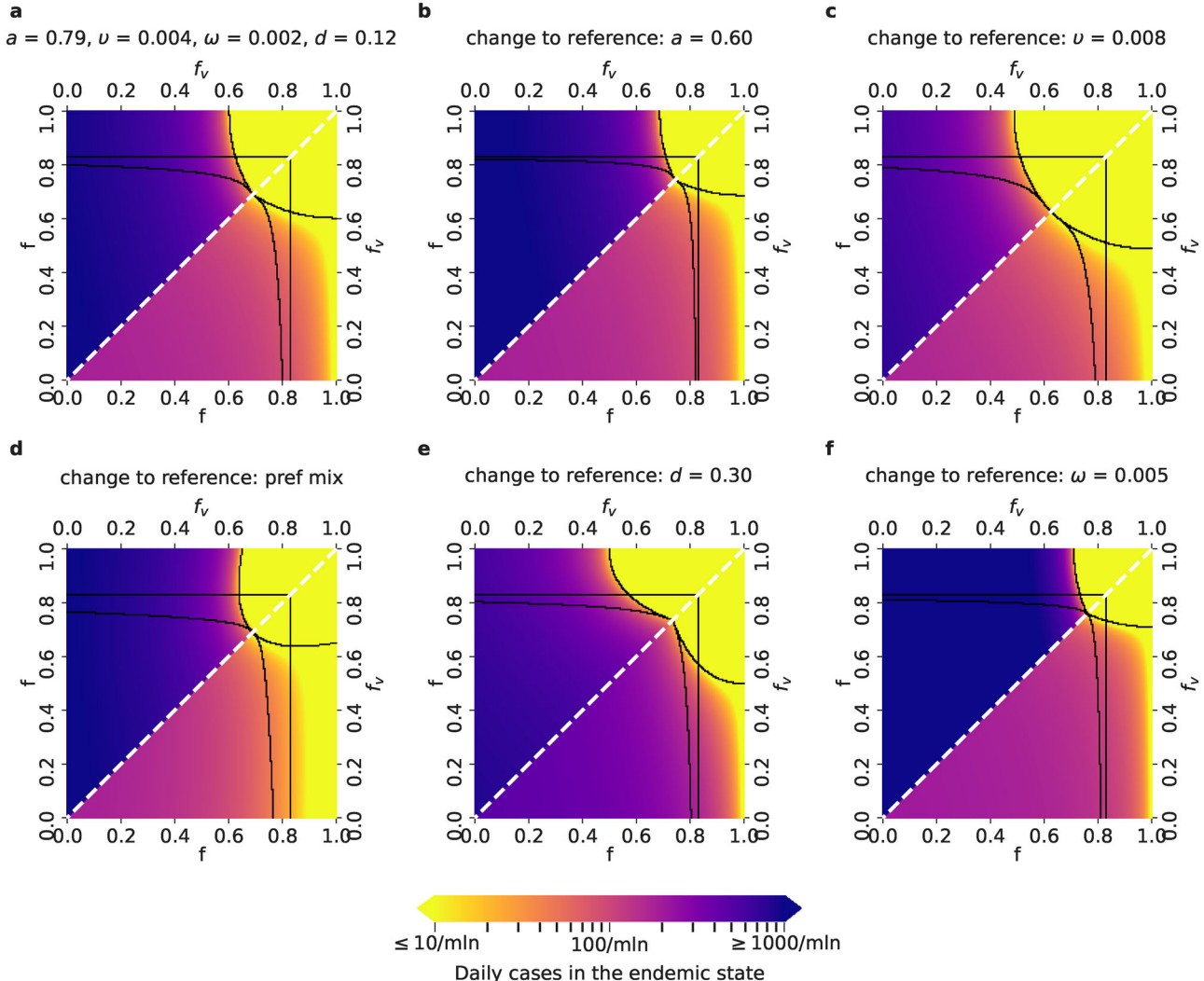

**Fig. 3 Daily COVID-19 infection cases in the endemic state for different parameter setups and the Delta variant.** Lower triangles show the daily infection numbers in the vaccinated, and upper triangles in the unvaccinated population in the endemic state of the epidemics, for the relevant $f - f_v$ parameter space, where $f_v \leq f$. The color scale spans from no more than 10 (yellow) up to 1000 and more daily cases per million people (dark violet). Parameter setups as well as the black borders that delimit the five regions are defined as in Fig. 2: **a** Reference setup, with $a = 0.79$ (corresponding to the effectiveness of the Comirnaty vaccine on the Delta variant), $v = v_r = 0.004$, $\omega = 0.002$, $d = 0.12$ (fraction of never-vaccinated in the United Kingdom) and proportional mixing. **b** Setup with a decreased vaccine effectiveness: $a = 0.6$ (corresponding to the effectiveness of the Vaxzevria vaccine on the Delta variant). **c** Setup with an increased vaccination rate: $v = v_r = 0.008$. **d** Setup with preferential (instead of proportional) mixing. **e** Setup with an increased fraction of people who will not get vaccinated: $d = 0.3$ (fraction of never-vaccinated in France). **f** Setup with an increased waning rate: $\omega = 1/200$.

population without immunity gained via vaccination requires long-term high restriction levels, and as such seems politically unfeasible.

When changing two parameters simultaneously in order to assess synergies, we find that a decreased vaccine effectiveness or an increased share of never vaccinated or an increased waning rate can barely be offset by an increase in vaccination speed. Both a decreased vaccine effectiveness and an increase in the share of never vaccinated in combination with an increased waning rate considerably increase the minimum restriction level that is adequate to ensure resurgence can be avoided. The latter (increased $d$, increased $\omega$ as compared to the reference) is the most pessimistic of the considered scenarios, with $f_{min} = 0.8$.

This analysis highlights the importance of vaccine effectiveness, vaccination speed, but also of the fraction of the never-vaccinated. Such demanding requirements for stringent minimum common restrictions could be reduced if novel vaccines with higher

effectiveness become available, if faster and wider vaccination programs are implemented, and finally, if the never-vaccinated fraction shrinks.

**Endemic state analysis reveals the possibility of large daily infection and hospitalization numbers.** For a given restriction combination **f**, the above analyzed asymptotic instantaneous reproduction number $\mathcal{R}^*$ (Fig. 2 and Supplementary Figs. S2, S3 and S4) indicates how quickly the infections grow shortly after the restrictions are set to **f** in the asymptotic state; however, it does not provide insight into the daily infection or hospitalization numbers the system converges to. To this end, we first compute the daily infection numbers both in the vaccinated and the unvaccinated subpopulations in the endemic state, as functions of the restrictions **f** for the Delta variant (Fig. 3) and compare it to the scenarios achieved with the Alpha variant (Supplementary Fig. S5). In contrast to the computation of the instantaneous reproduction number $\mathcal{R}^*$ and its asymptotic values, which is

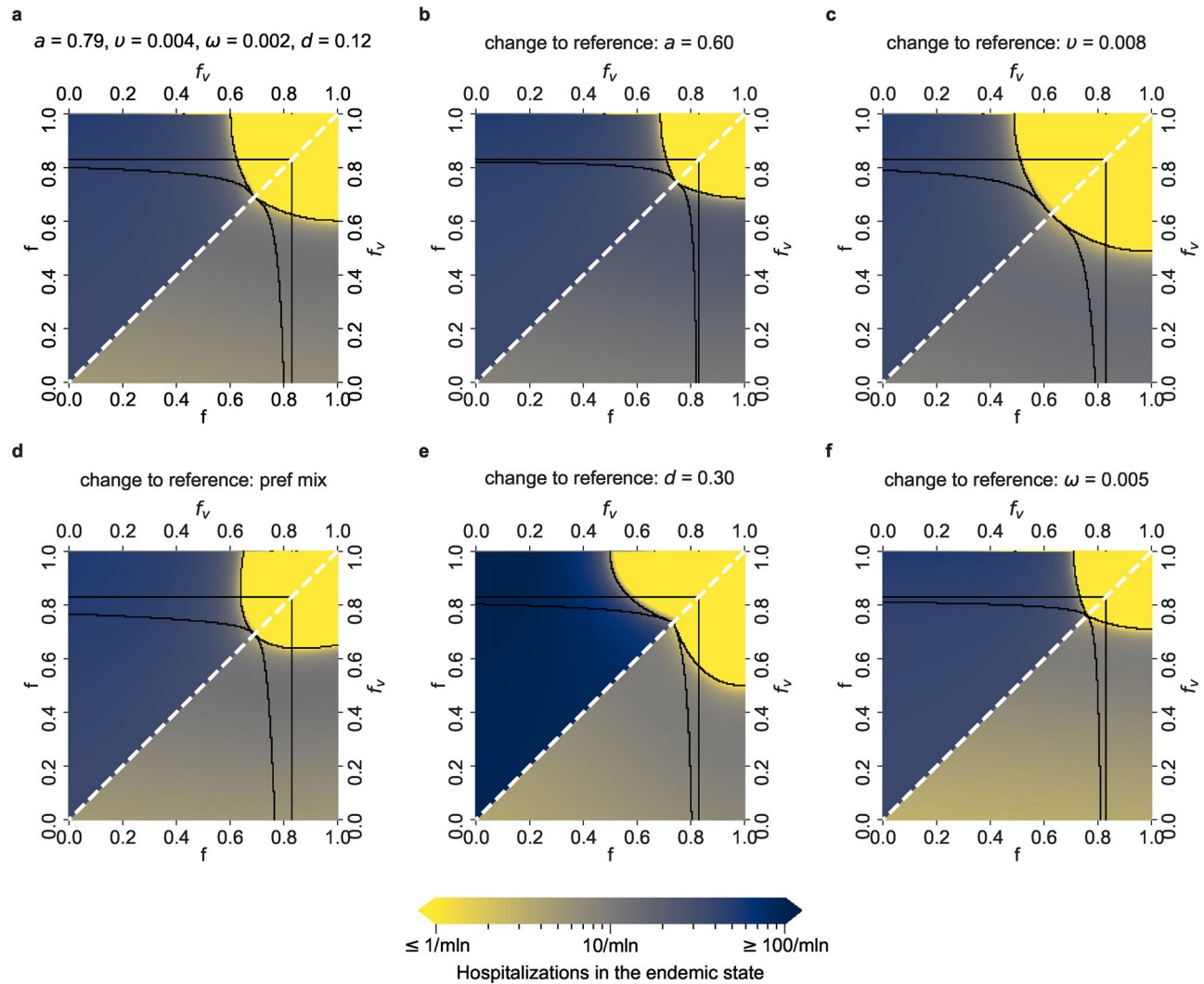

**Fig. 4 Daily COVID-19 hospitalized cases in the endemic state for different parameter setups and the Delta variant.** Lower triangles show the daily hospitalized numbers in the vaccinated population, and upper triangles in the unvaccinated population, in the endemic state of the epidemic, for the relevant $f - f_v$ parameter space, where $f_v \leq f$. The color scale spans from no more than 1 (yellow) up to 100 and more daily hospitalized cases per million people (navy blue). Parameter setups as well as the black borders that delimit the five regions are defined as in Fig. 2: **a** Reference setup, with $a = 0.79$ (corresponding to the effectiveness of the Comirnaty vaccine on the Delta variant), $v = v_r = 0.004$, $\omega = 0.002$, $d = 0.12$ (fraction of never-vaccinated in the United Kingdom) and proportional mixing. **b** Setup with a decreased vaccine effectiveness: $a = 0.6$ (corresponding to the effectiveness of the Vaxzevria vaccine on the Delta variant). **c** Setup with an increased vaccination rate: $v = v_r = 0.008$. **d** Setup with preferential (instead of proportional) mixing. **e** Setup with an increased fraction of people who will not get vaccinated: $d = 0.3$ (fraction of never-vaccinated in France). **f** Setup with an increased waning rate: $\omega = 1/200$.

based on the analysis of the linearized system of the ordinary equations in the VAP-SIRS model, the endemic state is based on the computation of the stationary point of the full set of the equations (Supplementary Note 1).

For all parameter setups, in all regions apart from the subcritical (−) region, the daily infections in the endemic state will exceed 10 per million, which is the tolerance threshold for efficient test, trace and isolation policy[58]. For the setups that correspond to low vaccination effectiveness or short waning time, the endemic state is most unfavorable, as the daily infections can exceed 1000 daily cases per million. A high (re-)vaccination rate is crucial to expand the safe region (Fig. 3c). A sharp transition can be seen between favorable and unfavorable parameter setups. In the endemic state, the daily infection numbers in the vaccinated subpopulation can exceed that of the unvaccinated subpopulation, which underlines the risks of waning immunity.

Considering the parameter setups that arise by changing two parameters at once with respect to the reference setup gives insights about their joint effects, shown in Supplementary Fig. S6; the effect of simultaneous parameter variations is akin to that described earlier for the values of asymptotic $\mathcal{R}^*$ and time to critical thresholds in Supplementary Fig. S3.

Again, comparison with the endemic infection numbers predicted for the Alpha variant (Supplementary Figs. S5 and S7) shows that Delta has considerably narrowed opportunities to reduce restrictions for the VP holders, underlining the negative impact of the higher transmissibility of the Delta variant and lower effectiveness of the vaccines on this variant.

Besides the computation of daily infection numbers, we compute daily hospitalization numbers for the Delta (Fig. 4) and Alpha (Supplementary Fig. S8) variants in the endemic state (see the Supplementary Note 1 for details). The benefit of vaccination in reducing hospitalizations is striking: for all regions

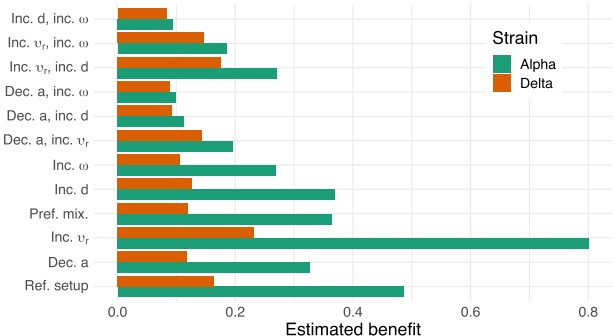

**Fig. 5 The estimated benefit of vaccination passes (VPs).** For each parameter setting (reference setup and single or double parameter value changes to it; y axis), the benefit of VPs was evaluated (x axis) for two SARS-CoV-2 strains: Alpha (green) and Delta (red).

apart from the (−) region, the number of hospitalized cases in the unvaccinated population is roughly an order of magnitude larger than in the vaccinated population. In addition, the unvaccinated population is even more prone to hospitalization when the fraction of never-vaccinated is larger (Fig. 4c and Supplementary Fig. S8c). This negative effect of the never-vaccinated population fraction is dominant also when double changes of parameter values are considered with respect to the reference setup, both for the Delta (Supplementary Fig. S9) and the Alpha variant (Supplementary Fig. S10). Again, the comparison between the Delta and the Alpha variant highlights the deleterious effect of increased transmissibility of the Delta variant. Clearly, to keep hospitalizations at the same level as for the Alpha variant, the Delta variant requires much stricter restrictions for the VP holders and the rest of the population.

**The benefit of VPs is larger for the Alpha than for the Delta variant, and strongly depends on the (re-)vaccination rates.** The above analysis demonstrates the potential risks of increased infection and hospitalization numbers. The additional freedom for VP holders can be considered beneficial for society as long as it does not lead to an uncontrolled surge of infections. Consequently, considering the relevant parameters as $f_v \leq f$, we estimate the benefit of VPs as a value in [0, 1] given by the fraction of the relevant $f - f_v$ parameter space where the asymptotic instantaneous reproduction number satisfies $\mathcal{R}^* \leq 1$. For example, for the Delta variant and the reference setup, the benefit of VPs can be obtained as a fraction of the corresponding upper triangle in Fig. 2, which is colored in white or shades of blue.

The estimated benefit of VPs for the reference parameter setup and the Alpha variant is almost 0.5 (Fig. 5). Increasing the (re-)vaccination rates has the most positive and the largest impact on the VP benefit, increasing it to around 0.8 for the Alpha variant. All other considered changes to the reference setup (decreasing the effectiveness of the vaccine, preferential instead of proportional mixing, increased waning rate $\omega$, increasing the fraction of never-vaccinated $d$) decrease the VP benefit. Strikingly, for half of the considered parameter setups (the reference setup and setups with single changes to it), the benefit is over two-fold reduced for the Delta variant, as compared to the Alpha variant. The maximum estimated benefit for the Delta variant is only around 0.23.

## Discussion

Introducing VPs is widely seen as a means to opening up economies and societies, despite the ongoing epidemic. A recent complication in this respect is the rise of the Delta variant with its higher transmissibility and decreased vaccine effectiveness. To inform this discussion, we extend a SIR model to reflect vaccination dynamics and possibly different restrictions for VP holders, with empirical parameters for both the Alpha and Delta variant.

VAP-SIRS deliberately keeps several aspects simple. The model is not compartmentalized for age groups and does not consider mortality or intensive care unit utilization like some other models, albeit in the context of exploring different parameters than larger freedom for VP holders[32,34,36–39]. In this context, the advantage of our model is that it is enriched in features such as revaccinations and waning immunity, which have been shown to be very relevant not only in the long term[22,27]. Avoiding another wave is a prudent goal due to the threats it poses, in the form of long-term health effects, the deleterious impact on societies and the emergence of new variants. Possible extensions to our model could include inter-individual variations in immunity, which would render it relevant for people with immunodeficiencies. The presented analysis has been performed assuming that without restrictions, the maximum reproduction number of the virus is $R_{max} = 6$ or $R_{max} = 4$ for Delta and Alpha variants, respectively. More transmissible variants could easily be modeled by fixing higher values of $R_{max}$. Possible future variants, for which existing vaccines may potentially be less effective could be considered using our model by fixing smaller vaccine effectiveness parameter $a$ than the values we considered. We also do not account for seasonality, which seems to have a dampening effect on epidemic dynamics during the summer months, when it is possible to temporarily reduce restrictions. Not all analyzed parameter values are exactly known, such as the post- vaccination or natural immunity waning time. We, however, fix optimistic values for such parameters, and show that unfavorable infection dynamics can still be obtained even under optimistic assumptions. Clearly, the assumption of a constant vaccination rate is—although common in the literature—a simplification of reality. For instance, booster shots are usually recommended only after a minimal time span of six months after the last vaccine dose, in order to maximize the effect on affinity maturation and immunological memory. Instead, our assumptions of a fixed vaccination rate corresponds to a situation where boosting shots could be taken earlier than this minimal time span, as is implemented in some countries facing another wave. Such minimal time delay could be accounted for using a delay equation, but we do not expect that such modification would affect the conclusions from our analysis. The presented analysis assumes that the initial number of infected and the number of recovered are both negligible. Given the large heterogeneity of infection dynamics[59], it is impossible to assign initial numbers that would be representative for all countries. We thus choose a universal reference assuming these numbers are very small or zero. The Shiny app for VAP-SIRS simulations, however, allows initializing with different values. For example, since the recovered, similarly to vaccinated, are also immunized for some time, the main conclusion when increasing the initial recovered number to some level is that it moves the dynamics forward in time in our analysis, as if the vaccinated sub-population started at this level.

Despite limitations, our model accounts for key parameters influencing infection dynamics and gives valuable insights into policies pertaining to the introduction of VPs, rendering the valid goal of avoiding resurgence attainable. We find that a wide range of the VAP-SIRS model parameter choices, even optimistic ones, show the possibility of an epidemic resurgence for both variants. The risk of resurgence is higher in the case of implemented VP, i.e., with VP holders enjoying reduced restrictions. The resurgence can be avoided in the short and in the long run only when the restrictions are kept high for the rest of the population, and the reduction for the VP holders is moderate or small, especially for the Delta variant. The main driver of this phenomenon is the

potential lack of immunity of VP holders. With a VP, people enjoy lower restrictions, while some actually remain both susceptible and potentially contagious because the vaccine was ineffective or the immunity has waned.

For all analyses, a comparison between values for the Alpha and Delta variants shows that Delta has drastically worsened all scenarios. Two illustrative findings are that: (i) the minimum level of common restrictions to avoid resurgence in the reference setup has doubled from 0.29 (Alpha) to 0.62 (Delta), and that (ii) the largest VP benefit has decreased almost four-fold from around 0.8 for Alpha to 0.23 for Delta.

Changing key parameters such as vaccine effectiveness, (re-)vaccination rate, or waning immunity rate to realistic levels found in studies or certain countries shows the expected effect these changes would have on infection dynamics. We quantified these effects by evaluating the times to overcriticality, asymptotic instantaneous reproduction number $\mathcal{R}^*$, minimum necessary common restriction level that avoids resurgence in the long term, numbers of cases per million in the endemic state, numbers of hospitalizations in the endemic state, and VP benefit for the relevant range of possible restrictions for the VP holders and the rest of the population. As expected, the model shows that there is a larger selection of admissible restrictions' combinations under high vaccine effectiveness, low share of never vaccinated, a higher (re-)vaccination rate, slowly waning immunity, and proportional social mixing. For the Delta variant, however, and even for optimistic parameter setups, the room for manoeuvre in terms of lowering the restrictions is very small. Moreover, not all of these parameters are amenable to policy action. In a nutshell, our results consistently suggest that with the Delta variant and with the way the vaccination program and introduction of VPs is currently implemented, unfavorable developments of the epidemic are likely, and to counteract these developments and to maximize possible freedoms for their citizens, decision makers should exploit all possibilities to enhance the development of effective vaccines, increase vaccination speed and the number of vaccinated.

It is noteworthy that VP holders are less likely to be tested, as they are assumed to be protected and they may exhibit milder symptoms. Therefore, their potential infection is more likely to remain undetected, resulting in an effect similar to that of lowering restrictions. To prevent undesirable outcomes, the testing and quarantine criteria should be applicable also to the VP holders. Testing should aim at detection of vaccinated people that have lost, or have never gained, immunity. Finally, temporary VPs could be considered, with their prolongation conditioned on high antibody level or recent (re-)vaccination.

The utilization of tools such as the VAP-SIRS model, along with different tools available to policymakers should be explored in the context of monitoring the implementation of VPs, including the EU DCC measures, to ensure optimization of key parameters. In this manner, evidence-informed policy making would be safeguarded as would the best possible outcomes in terms of effectively combating the current pandemic.

## Data availability
All source data underlying the graphs and charts presented in the main and supplementary figures are publicly available on-line in the GitHub repository: https://github.com/eMaerthin/VAP_SIRS_Analysis as text (.csv) files.

## Code availability
The VAP-SIRS model was implemented using R version 4.0.2 along with the shiny package to build an interactive web application that allows to simulate the model. The code of the model is available on-line in the GitHub repository: https://github.com/storaged/VAP-SIRS[60], and the on-line tool is available at http://bioputer.mimuw.edu.pl:85/VAP-SIRS/. The code to generate Figures from the main article and supplementary information is available at https://github.com/eMaerthin/VAP_SIRS_Analysis.

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

## Acknowledgements

SC acknowledges support by University of Malta. EP acknowledges support by the University of Crete. TC has received funding from the European Union's Horizon 2020 research and innovation program under grant agreement No 101016233 (PERISCOPE). GG acknowledges support by the University of Trento within the COVID-19 Strategic Project MOSES (Models for Reasoning about the Spreading of Diseases). MP was supported by the Slovenian Research Agency (Grant Nos. P1-0403 and J1-2457).

## Authors contributions

A.G., K.G., T.K. and E.S. conceived the VAP-SIRS model—with input and feedback on the model and results from T.C., G.G., M.P., E.P. and M.R. T.K. performed the stability analysis. K.G. implemented model simulations and the Shiny application for visualizations. M.B. implemented the stability analysis. S.C., T.C., E.P. and M.R. performed literature search. E.S. supervised the study. All authors wrote and provided critical feedback to the manuscript.

## Competing interests

Other projects in the research lab of E.S. are co-funded by Merck Healthcare KGaA. The remaining authors declare no competing interests.
