## [Peer Review File · Communications Medicine]

Reviewers' comments:

Reviewer #1 (Remarks to the Author):

The authors study the long term dynamics of COVID-19 variants in the presence of immunity waning, (re-)vaccination, and nonpharmaceutical measures that reduce social contacts and may be different for vaccinated and unvaccinated populations. In particular, the authors wish to assess the impact of vaccination passes (VPs) in the dynamics of the disease.

The approach is based on a simplified compartmental model based on SIRS dynamics, extended to include other compartments needed to describe partially effective, non-universal (re-)vaccination, and immunity waning. The model has a large number of parameters, whose range of values are taken from the literature, when available, or fixed according to clearly stated assumptions. The focus of the analysis is the impact of the parameters f_v and f that measure the restrictions imposed on VP holders and on the general population.

The paper is clear and well written, and it offers a thorough exploration of long term COVID-19 scenarios, with emphasis on the role of VPs. While the methodology employed is well known, I believe the results are interesting and new.

Regarding the robustness of the conclusions and their relevance to inform policy making:

1. My main concern is the assumption that at the beginning of the vaccination the number of infected and the number of recovered are both negligible. A more realistic value for the initial fraction of recovered in the population should be taken both in the long term integration of the different set ups and in the stability analysis. This will change the quantitative aspects of the predictions and may change some of the qualitative aspects as well.
2. It would be useful to translate the results of Figures 4 and 5 in terms of hospitalizations, or severe disease cases.
3. The random revaccination assumption is also clearly unrealistic because booster shots will almost surely be scheduled according to the initial immunization calendar. I think this should at least be discussed.
4. Finally, one may argue that the use of VPs and the value of the fraction d of people who will never get vaccinated are not uncorrelated. This aspect could also be discussed in the framework of the results of the manuscript.

The following minor aspects may deserve the authors' attention:

5. There are lots of 'R's, with different meanings, I suggest changing all those that have to do with reproduction numbers to calligraphic fonts.
6. The supplementary material is using different notation from the main text for the basic equations. Moreover, it does not offer any analysis of the endemic state, it merely says that it is an equilibrium.
7. The preferential mixing introduced in page 21, lines 466-467 is not obvious to me, I think a little

more explanation would be useful.

8. Line 469 of the manuscript, I do not think 'simulations' is the right word for the section title.

9. In line 488, why talk about 'the largest and the second largest eigenvalues' if there are only two eigenvalues.

10. I fail to see the point of the whole digression through stochastic processes in pages 24-25. I think it adds nothing to the definition of the instantaneous reproduction number in the framework of ODEs as in lines 498-499.

Reviewer #2 (Remarks to the Author):

Review: Assessing the risk of COVID-19 epidemic resurgence in relation to the Delta variant and to vaccination passes.

In this paper, the authors develop a novel epidemic model for investigating the impact of vaccination and vaccine passes at controlling future outbreaks. This is an interesting concept, addressing an important and timely question, and the model is thoroughly analysed. I have no major concerns with the paper, but have a few small areas that could do with addressing.

Firstly, although not the primary aim of the paper, I would be interested in seeing an evaluation of the benefit of vaccine passes. The authors argue that passes can be useful since they allow freedom to be increased, whilst maintaining some infection control. It would be interesting to therefore assess the impact of vaccine passes at leading to more control, for a given level of freedom. I am not sure what the best way to do this might be, but it could be worth selecting a given value of population level freedom (i.e. averaged across vaccinated and unvaccinated), and showing that having vaccinated individuals with higher freedom (and therefore lower freedom in unvaccinated) increases control of the epidemic. Alternatively, some other measure of the benefit of vaccine passes might be suitable. This is not essential for acceptance, just something that would make an interesting addition to the paper.

Line 170 – What is meant here by “composition of the population”. Is this the distribution of individuals across susceptible classes by vaccine status? Are both I and R assumed small or just I assumed small? This paragraph could do with rephrasing to clarify what is being assumed here.

Line 51 – “parental strain”. I think a better phrase would be more appropriate here. Maybe “previously dominant strains”.

Line 56 – typo “still highly effect at”.

Line 84 – I think it would be worthwhile to define here what each of the letters in the “VAP-SIRS” model stand for.

Line 181 – typo. “evolution fo R*”.

Double check for typos throughout.

We are very grateful for all the received comments. The detailed answers to all the Reviewers' comments are written in blue font below. A version of the manuscript with all changes highlighted in magenta is attached. We believe the revision has greatly improved the manuscript.

Reviewer I

The authors study the long term dynamics of COVID-19 variants in the presence of immunity waning, (re)-vaccination, and nonpharmaceutical measures that reduce social contacts and may be different for vaccinated and unvaccinated populations. In particular, the authors wish to assess the impact of vaccination passes (VPs) in the dynamics of the disease.

The approach is based on a simplified compartmental model based on SIRS dynamics, extended to include other compartments needed to describe partially effective, non-universal (re-)vaccination, and immunity waning. The model has a large number of parameters, whose range of values are taken from the literature, when available, or fixed according to clearly stated assumptions. The focus of the analysis is the impact of the parameters f_v and f that measure the restrictions imposed on VP holders and on the general population.

The paper is clear and well written, and it offers a thorough exploration of long term COVID-19 scenarios, with emphasis on the role of VPs. While the methodology employed is well known, I believe the results are interesting and new.

Regarding the robustness of the conclusions and their relevance to inform policy making:

Question 1. *My main concern is the assumption that at the beginning of the vaccination the number of infected and the number of recovered are both negligible. A more realistic value for the initial fraction of recovered in the population should be taken both in the long term integration of the different set ups and in the stability analysis. This will change the quantitative aspects of the predictions and may change some of the qualitative aspects as well.*

Answer I.1. Thank you for highlighting this point. Indeed, the initial number of people infected or recovered in the population (R) is non-zero in real life and, thanks to natural immunization, this should impact the dynamics of the epidemic, by postponing the potential peak of daily cases. At the same time, however, as we look at the data, the heterogeneity across the world in the number of infected or recovered at certain time points is large¹. It is impossible to assign numbers representative for all countries. Therefore, zero was a natural reference value for the number of infected and recovered in the results presented in the manuscript.

However, the shiny-app attached to the manuscript (available at <http://bioputer.mimuw.edu.pl:85/VAP-SIRS/>) enables the user to investigate the changes in the epidemic evolution when the initial value of infected I or recovered R is non-zero. For example, since the recovered, similarly to vaccinated, are also immunized for some time, the main conclusion when increasing the initial recovered number to some level is that it moves the dynamics forward in time in our analysis, as if the vaccinated sub-population started at this level.

This limitation of our analysis is now covered in the Discussion page X paragraph Y. **[TODO add the page and paragraph numbers once all agree with the final text]**

Q2. *It would be useful to translate the results of Figures 4 and 5 in terms of hospitalizations, or severe disease cases.*

¹ Iftekhhar, E. N., Priesemann, V., Balling, R., et al. (2021). *A look into the future of the COVID-19 pandemic in Europe: an expert consultation*. **The Lancet Regional Health - Europe** (Vol. 8, p. 100185). Elsevier BV. <https://doi.org/10.1016/j.lanep.2021.100185>

Thank you for this inspiring suggestion. We have extended the model to be able to estimate the daily number of hospitalizations, both in the vaccinated and in the unvaccinated subpopulations, in the endemic state. We describe the extension in the Supplementary Text. Four new Figures presenting the hospitalized numbers for different parameter settings and different restrictions were added, one in the main text (Figure 4), and three in the Supplement (Supplementary Fig. SX--SX) . Conclusions from this new analysis are described on page X. Consequently, we have removed the description of the limitation of the model that we do not consider features such as hospitalizations from the Discussion.

Q 3. *The random revaccination assumption is also clearly unrealistic because booster shots will almost surely be scheduled according to the initial immunization calendar. I think this should at least be discussed.*

A I.3. Indeed, the model's assumption of a constant vaccination rate is - although common in the literature - a simplification of reality. For instance, as pointed out by the Reviewer, booster shots are usually recommended and allowed only after a minimal time span of 6 months after the last vaccine dose. Instead, the assumptions of a fixed vaccination rate made in our model corresponds to a situation where boosting shots could be taken earlier than this minimal time span. To account for such minimal time constraints the model could be modified by including a delay equation and additional parameters, but we do not expect that such modification would affect the conclusions from our analysis. This is now discussed on page X.

Q 4. *Finally, one may argue that the use of VPs and the value of the fraction d of people who will never get vaccinated are not uncorrelated. This aspect could also be discussed in the framework of the results of the manuscript.*

A I.4. Thank you for raising this interesting point. Indeed, it is likely that with higher freedom offered for the VP holders less people will stay in the never vaccinated population. In such a way, we could expect a negative correlation between f_V and d . We however did not account for this in the VAP-SIRS model and we discussed it as an interesting future research direction on page X line Y.

The following minor aspects may deserve the authors' attention:

Q 5. *There are lots of 'R's, with different meanings, I suggest changing all those that have to do with reproduction numbers to calligraphic fonts.*

A I.5. We applied the changes as suggested by the Reviewer: all reproduction numbers are now denoted as calligraphic R.

Q 6. *The supplementary material is using different notation from the main text for the basic equations. Moreover, it does not offer any analysis of the endemic state, it merely says that it is an equilibrium.*

A I.6. In the supplementary materials the missing subindices were corrected, also some parts were reformulated. In the main text, we now clarify that the endemic state is computed (not analysed) in the Supplementary Text.

Q 7. *The preferential mixing introduced in page 21, lines 466-467 is not obvious to me, I think a little more explanation would be useful.*

A I.7. Thank you for this comment. To improve the explanation and give the literature context, we have added the following two sentences:

'Preferential mixing is a common, socio-psychological motivated mixing scheme alternative to proportional

mixing. In this scheme the group interaction is still proportional, but biased by the relative degree of freedom given to the passport holders. Preferential mixing as a modulation of proportional mixing was previously studied in the context of infectious diseases by Glasser et al².

Q 8. *Line 469 of the manuscript, I do not think 'simulations' is the right word for the section title.*

A I.8. The section title was changed to: 'Numerical integration and parameter values'.

Q 9. *In line 488, why talk about 'the largest and the second largest eigenvalues' if there are only two eigenvalues.*

A I.9. The line was rephrased to: Since both the eigenvalues λ_{max} and $\lambda_2 \leq \lambda_{max}$ of J_{sub} are real.

Q 10. *I fail to see the point of the whole digression through stochastic processes in pages 24-25. I think it adds nothing to the definition of the instantaneous reproduction number in the framework of ODEs as in lines 498-499.*

A I.10. We added this part to better explain why we call this reproduction number an instantaneous reproduction number, and to show the derivation of the doubling time for this system. We added a motivation sentence before this part (page X). Describing the corresponding branching process allows us to also show how the doubling time would have been computed in case of more complex systems than SIRS, such as SEIRS. Only a small change is needed in the derivation to extend to those systems. A comment on this aspect was added on page X, paragraph Y. We find this part also useful for didactic reasons. Still, we can remove it if the Reviewer still finds it out of place.

² J. Glasser, Z. Feng, A. Moylan, S. Del Valle, and C. Castillo-Chavez (2012) *Mixing in age-structured population models of infectious diseases*, **Mathematical Biosciences**, (vol. 235, no. 1, pp. 1–7.) Elsevier BV. [doi: 10.1016/j.mbs.2011.10.001](https://doi.org/10.1016/j.mbs.2011.10.001).

Reviewer II

In this paper, the authors develop a novel epidemic model for investigating the impact of vaccination and vaccine passes at controlling future outbreaks. This is an interesting concept, addressing an important and timely question, and the model is thoroughly analysed. I have no major concerns with the paper, but have a few small areas that could do with addressing.

Q 1. *Firstly, although not the primary aim of the paper, I would be interested in seeing an evaluation of the benefit of vaccine passes. The authors argue that passes can be useful since they allow freedom to be increased, whilst maintaining some infection control. It would be interesting to therefore assess the impact of vaccine passes at leading to more control, for a given level of freedom. I am not sure what the best way to do this might be, but it could be worth selecting a given value of population level freedom (i.e. averaged across vaccinated and unvaccinated), and showing that having vaccinated individuals with higher freedom (and therefore lower freedom in unvaccinated) increases control of the epidemic. Alternatively, some other measure of the benefit of vaccine passes might be suitable. This is not essential for acceptance, just something that would make an interesting addition to the paper.*

A II.1 Thank you for this great idea! We have implemented the suggestion in the revised manuscript. As correctly stated by the reviewer, the VPs can be useful since they allow freedom to be increased, whilst maintaining some infection control. Taking this into account, we estimate the benefit of VPs as the fraction of the relevant $f - f_v$ parameter subspace, for which the asymptotic instantaneous reproduction number is not larger than 1. The computed benefit is featured in a new Fig. 6. An explanation and results of the evaluation are described in a new subsection 'The benefit of VPs is larger for the Alpha than for the Delta strain, and strongly depends on the (re-)vaccination rates', page X. **[todo add correct page numbers]**

Q 2. *Line 170 – What is meant here by “composition of the population”. Is this the distribution of individuals across susceptible classes by vaccine status? Are both I and R assumed small or just I assumed small? This paragraph could do with rephrasing to clarify what is being assumed here.*

A II.2 By “composition of the population” we mean here the population fractions of the groups: susceptible with VPs (V), susceptible without VPs (S), and the immune group V. We now explain it at line X on page X.

Q 3. *Line 51 – “parental strain”. I think a better phrase would be more appropriate here. Maybe “previously dominant strains”.*

A II.3 Rephrased as suggested.

Q 4. *Line 56 – typo “still highly effect at”.*

A II.4 The phrase was reformulated to: are effective in preventing.

Q 5. *Line 84 – I think it would be worthwhile to define here what each of the letters in the “VAP-SIRS” model stand for.*

A II.5 In the mentioned line we explain the abbreviation as: VAccination Passes in Susceptible-Infectious-Recovered-Susceptible model.

Q 6. *Line 181 – typo. “evolution fo R*”.*

A II.6 The typo was corrected.

Q 7. *Double check for typos throughout.*

A II.7 The full text was proofread by all co-authors and, hopefully, is free of typos.

REVIEWERS' COMMENTS:

Reviewer #1 (Remarks to the Author):

The authors have addressed most of my concerns in the revised version and given arguments justifying their approach in the rebuttal letter.

It is a bit annoying that the latter was submitted still in draft form, and that pages and paragraph numbers of the changes under discussion are still missing. However, given the importance of timely contributions on the subject, I do not think that publication should be delayed because of this.

Reviewer #2 (Remarks to the Author):

The authors have taken into account all of my comments and concerns with the initial submission. I appreciate the addition of the estimated benefit calculation, and found this interesting to read.

I am happy to approve this submission. However, there is one thing that would be worth amending. The authors switch between Fig. X and Figure X multiple times without the paper. It would be worth making this consistent, as it would make it easier for the reader to search through the manuscript.